# COUNTERFACTUAL FAIRNESS FROM PARTIALLY DAGS: A GENERAL MIN-MAX OPTIMIZATION FRAMEWORK

## ABSTRACT

Developing fair automated machine learning algorithms is critical in making safe and trustworthy decisions. Many causality-based fairness notions have been proposed to address the above issues by quantifying the causal connections between sensitive attributes and decisions, and when the true causal graph is fully known, certain algorithms that achieve counterfactual fairness have been proposed. However, when the true causal graph is unknown, it is still challenging to effectively and well exploit partially directed acyclic graphs (PDAGs) to achieve counterfactual fairness. To tackle the above issue, a recent work suggests using non-descendants of sensitive attribute for fair prediction. Interestingly, in this paper, we show it is actually possible to achieve counterfactual fairness even using the descendants of the sensitive attribute for prediction, by carefully control the possible counterfactual effects of the sensitive attribute. We propose a general min-max optimization framework that can effectively achieve counterfactual fairness with promising prediction accuracy, and can be extended to maximally oriented PDAGs (MPDAGs) with added background knowledge. Specifically, we first estimate all possible counterfactual treatment effects of sensitive attribute on a given prediction model from all possible adjustment sets of sensitive attributes. Next, we propose to alternatively update the prediction model and the corresponding possible estimated causal effects, where the prediction model is trained via a min-max loss to control the worst-case fairness violations. Extensive experiments on synthetic and real-world datasets verifying the effectiveness of our methods.

## 1 INTRODUCTION

Making automated machine learning algorithms fair is critical to producing safe and trustworthy decisions for subgroups or individuals with different sensitive attributes (e.g., gender and race) (Brennan et al., 2009; Dieterich et al., 2016; Hoffman et al., 2018; Chouldechova et al., 2018). To achieve fair predictions, association-based and causality-based fairness notions have been proposed. Specifically, association-based fairness investigates the statistical independence between sensitive attributes and predicted outcomes (Chouldechova, 2017; Dwork et al., 2012; Hardt et al., 2016), whereas causality-based subgroup fairness constrains the causal effect of sensitive attributes on predicted outcomes (Zhang and Bareinboim, 2018; Zhang et al., 2017a;b; 2018a;b).

Among the above fairness notions, counterfactual fairness (Kusner et al., 2017; Chiappa, 2019; Nabi and Shpitser, 2018; Wu et al., 2019b) considers causal effects within particular individuals or groups, requiring that the predicted outcomes be the same across the real-world without intervention and the counterfactual world with intervention on sensitive attributes. Despite many algorithms have been developed to achieve counterfactual fairness, most of them require the true causal directed acyclic graph (DAG) is fully known. Nevertheless, true causal DAGs and structural equations are usually not directly available in practice. Moreover, without strong assumptions, e.g., linearity (Shimizu et al., 2006) and additive noise (Hoyer et al., 2008; Peters et al., 2014), the true causal DAG may not be recoverable from only the observed data, which raises a great challenge to achieve counterfactual fairness based on partially directed acyclic graphs (PDAGs).

To tackle the above problem, a recent work (Zuo et al., 2022) proposes to use observed data to first classify variables into three categories: *definite non-descendants*, *possible descendants*, and *definite descendants* of the sensitive attributes. Next, by noting that a prediction model would be counterfactually fair if the prediction model is a function of the non-descendants of sensitive

Table 1: Comparison of methods to achieve counterfactual fairness from PDAGs. Both FAIR and FAIRRELAX employ a two-stage approach: they first learn a CPDAG from observed data, and then make prediction with the definite non-descendants (and possible descendants) of the sensitive attribute. Our method alternatively updates the predictions using *all* variables and possible counterfactual treatment effects via a min-max optimization.

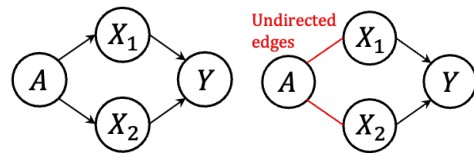

(a) A sample true DAG.    (b) CPDAG of (a).

Figure 1: A toy example for illustration: FAIR has *no available variables* for prediction; FAIRRELAX uses $\{X_1, X_2\}$ *without* further fairness constraint; OURS uses $\{A, X_1, X_2\}$ *with* a min-max constraint bounding all possible counterfactual treatment effect.

| Variable types | FAIR | FAIRRELAX | OURS |
|---|---|---|---|
| Definite non-descendants | ✓ | ✓ | ✓ |
| Possible descendants | × | ✓ | ✓ |
| Definite descendants | × | × | ✓ |

attributes (Kusner et al., 2017), two algorithms are proposed to achieve counterfactual fairness as shown in Table 1: FAIR, which makes predictions using *definite non-descendants*, and FAIRRELAX, which further incorporates *possible descendants* to make predictions.

Despite being theoretically sound, as shown in Table 1, both FAIR and FAIRRELAX forbid all definite descendants during the prediction model training, which results in very few attributes available for making prediction, which may significantly decreasing the accuracy. Especially, the sensitive attribute is usually an inherent nature of data hence many attributes are its descendants (Wu et al., 2019a).

We proceed with a toy example for illustration: Figure 1(a) shows a sampled DAG as the ground-truth, and given the observed data, FAIR and FAIRRELAX algorithms first learn a Markov equivalence class of DAGs that encode the same set of conditional independencies from the data, also known as a completely partially directed acyclic graph (CPDAG), as shown in Figure 1(b). One on hand, the *definite non-descendants* of sensitive attribute $A$ is a empty set, thus FAIR is unable to give valid predictions. On the other hand, the *possible descendants* of the sensitive attribute $A$ are $\{X_1, X_2\}$, thus FAIRRELAX uses both $X_1$ and $X_2$ to predict $Y$ by minimizing the empirical risk *without imposing any further fairness constraint*. However, such relaxation would lead to a serious violation of counterfactual fairness, due to the nodes used ($X_1$ and $X_2$ in this example) for outcome regression might be descendants of the sensitive attribute $A$ in the true DAG, as in Figure 1(a).

In this paper, we propose a general min-max optimization framework to achieve counterfactual fairness. Different from the previous variable selection-based methods, we exploit all variables to ensure relatively high prediction accuracy as in Table 1. Specifically, we first estimate all possible counterfactual treatment effects of sensitive attribute on predictions for a given prediction model. To estimate all possible causal effects, instead of enumerating all possible DAGs in the learned Markov equivalence class, inspired by the IDA framework (Maathuis et al., 2009), we propose a local algorithm to obtain possible adjustment sets of the sensitive attributes. Next, we propose to alternatively update the prediction model and the corresponding estimation of the possible causal effects, where the prediction model is trained via a min-max loss to control the worst-case fairness violations. Meanwhile, we show the proposed framework can be extended to maximally oriented PDAGs (MPDAGs) with added background knowledge.

The main contributions of this paper are:

- We propose a general min-max optimization framework to achieve counterfactual fairness, which enables to use all variables to achieve relatively high prediction accuracy, and can be extended to MPDAGs with added background knowledge.

- Based on the proposed framework, we provide an efficient algorithm to estimate all possible causal effects of sensitive attribute on predictions for MPDAGs.

- We further provide a joint learning approach that alternatively updates the prediction model and the corresponding estimation of the possible causal effects, where the prediction model is trained via a min-max loss to control the worst-case fairness violations.

- We conduct extensive experiments on synthetic and real-world datasets to demonstrate the effectiveness of our methods in achieving counterfactual fairness with promising accuracy.

## 2 PRELIMINARIES

### 2.1 DAGs, PDAGs, CPDAGs, AND MPDAGs

In a graph $\mathcal{G} = (V, E)$, where $V$ and $E$ represent the node set and edge set in $\mathcal{G}$, we say $\mathcal{G}$ is *directed*, *undirected*, or *partially directed* if all edges in the graph are directed, undirected, or a mixture of directed and undirected edges, respectively. The *skeleton* of $\mathcal{G}$ is an undirected graph obtained by removing all arrowheads from $\mathcal{G}$. Given a graph $\mathcal{G}$, an $X_i$ is called a *parent* of $X_j$ and $X_j$ is called a *child* of $X_i$ if $X_i \rightarrow X_j$ in $\mathcal{G}$. Also, $X_i$ is a *sibling* of $X_j$ if $X_i - X_j$ in $\mathcal{G}$. If $X_i$ and $X_j$ are connected by an edge, they are *adjacent*. The notation $pa(X_i, \mathcal{G})$, $ch(X_i, \mathcal{G})$, $sib(X_i, \mathcal{G})$, and $adj(X_i, \mathcal{G})$ respectively represent sets of parents, children, siblings, and adjacent vertices of $X_i$ in $\mathcal{G}$. A graph is termed *complete* if all distinct vertices are adjacent. A path is a sequence of distinct vertices $(X_{k_1}, \cdots, X_{k_j})$ where any two consecutive vertices are adjacent. If all distinct vertices in a graph are connected by a path, then the graph is *connected*. A path is called *partially directed* from $X_{k_1}$ to $X_{k_j}$ if $X_{k_i} \leftarrow X_{k_{i+1}}$ does not occur in $\mathcal{G}$ for any $i = 1, \ldots, j - 1$. If all edges on the path are directed (undirected), then the partially directed path is *directed* (respectively *undirected*).

In a directed acyclic graph (DAG), all edges are directed and there is no directed cycle. A partially directed acyclic graph (PDAG) may contain both directed and undirected edges without directed cycles. Two DAGs are Markov *equivalent* if they induce the same set of conditional independence relations (Pearl, 1988). A *Markov equivalence class*, denoted by $[\mathcal{G}]$, contains all DAGs equivalent to $\mathcal{G}$. A Markov equivalence class can be uniquely represented by a partially directed graph called *completely partially directed acyclic graph* (CPDAG) $\mathcal{G}^*$, in which two vertices are adjacent if and only if they are adjacent in $\mathcal{G}$, and a directed edge occurs if and only if it appears in all DAGs in $[\mathcal{G}]$ (Chickering, 2002a). Given explicit knowledge of some causal relationships between variables, or some model restrictions, one can obtain a refinement of this class, uniquely represented by a maximally oriented partially directed acyclic graphs (maximal PDAGs or MPDAGs).

### 2.2 COUNTERFACTUAL INFERENCE

We follow Pearl (2000) to define the structural causal model (SCM) as a triplet $(U, V, F)$ to describe the causal relationships between variables. Specifically, $V$ is a set of observable endogenous variables and $U$ is a set of latent independent background variables that cannot be caused by any variable in $V$. $F$ is a set of functions $\{f_1, \ldots, f_{|V|}\}$, one for each $V_i \in V$, such that $V_i = f_i(pa_i, U_i)$, where $pa_i \subseteq V \setminus \{V_i\}$ and $U_i \in U$. Notably, the set of equations $F$ induces a directed graph over the variables, here assumed to be a DAG, where the directed causes of $V_i$ represents its parent set in the causal graph. Given a distribution $P(U)$ over the background variables $U$, an intervention on variable $V_i$ is defined as the substitution of equation $V_i = f_i(pa_i, U_i)$ with the equation $V_i = v$ for some $v$.

Based on SCM, counterfactual inference aims to answer counterfactual questions in the counterfactual world. For example, in the context of fairness, let $A$, $\hat{Y}$, and $X$ denote sensitive attributes, decision-making on that individual, and other observable attributes, respectively. For an individual with background variables $U = u$, and observable variables $A = a$, $\hat{Y} = y$, and $X = x$, the counterfactual problem is formulated as "what would the value of $\hat{Y}$ be had $A$ taken another value $a'$", formally denoted as $\hat{Y}_{A \leftarrow a}(U)$. To solve the above counterfactual problem, counterfactual inference consists of the following three steps: *Abduction*, *Action*, and *Prediction*, as explained in more details in Chapter 4 of (Pearl et al., 2016) and Chapter 7.1 of (Pearl, 2000).

### 2.3 COUNTERFACTUAL FAIRNESS

Counterfactual fairness is a fairness criterion based on SCMs, which can be examined by using the aforementioned counterfactual inference. Let $A$, $Y$, and $X$ denote sensitive attributes, outcomes of interest, and other observable attributes, and $\hat{Y}$ be a predictor produced by a machine learning algorithm as a prediction of $Y$. We say $\hat{Y}$ is counterfactually fair towards an individual if it is the same in (a) the factual world and (b) a counterfactual world where the individual belonged to a different demographic group, i.e., the counterfactual treatment effect is zero (Mitchell et al., 2021).

**Definition 2.1** (Counterfactual fairness). *Predictor $\hat{Y}$ is counterfactually fair if under any context $X = x$ and $A = a$, we have*

$$P\left(\hat{Y}_{A \leftarrow a}(U) = y \mid X = x, A = a\right) = P\left(\hat{Y}_{A \leftarrow a'}(U) = y \mid X = x, A = a\right),$$

*for all $y$ and for any value $a'$ attainable by $A$.*

## 3 A GENERAL MIN-MAX OPTIMIZATION FRAMEWORK

### 3.1 MOTIVATION

When the true causal graph is unknown, to the best of our knowledge, Zuo et al. (2022) performed the first work to obtain a counterfactual fairness predictor on a MPDAG, which focuses on utilizing the properties of the causal graph (Level 1 in (Kusner et al., 2017)) – to make predictions with the definite non-descendants (and possible descendants) of the sensitive attribute, as shown in Table 1. However, further incorporating the descendants of sensitive attribute into the predictor may also achieve counterfactual fairness by "cancelling out" the counterfactual treatment effects, which utilizes the observed variables more sufficiently and further improves the accuracy of the prediction.

We provide an intuition for the rationality and the advantages of using all variables by adopting the toy example in Figure 1. Suppose the structural equations in Figure 1(a) are: $A = U_A$, $X_1 = A + U_1$, $X_2 = A + U_2$, and $Y = 2X_1 + X_2 + U_Y$, which satisfies the faithfulness assumption (Uhler et al., 2013). In such a case, as discussed before, the FAIR algorithm proposed in Zuo et al. (2022) prevents all variables from predicting $Y$, while the FAIRRELAX algorithm uses both $X_1$ and $X_2$ to predict $Y$ without imposing any fairness constraints, and therefore cannot achieve counterfactual unfairness, since $X_1$ and $X_2$ are descendants of $A$. To achieve more accurate predictions with counterfactual fairness guarantees, one may notice that a function of $X_1 - X_2$ can be used to predict $Y$. On the one hand, this is strictly counterfactually fair due to the fact that $X_1 - X_2 = U_1 - U_2$, which is independent of the sensitive attribute $A$. On the other hand, this is informative for predicting $Y$ due to $\mathrm{Cov}(X_1 - X_2, Y) = 2\,\mathrm{Var}(U_1) - \mathrm{Var}(U_2) \neq 0$.

However, the true DAG and the corresponding structural equations are unknown in many real-world scenarios, which poses a great challenge to estimate the possible counterfactual treatment effects. To address this problem, an intuitive approach is to first find a Markov equivalence class over all vertices, which can be achieved using standard causal discovery methods, e.g., PC (Spirtes et al., 2000) and GES (Chickering, 2002b), and then to globally enumerate all the possible DAGs in the equivalence class and estimate their causal effects for each. However, as discussed in Section 7 of (Zuo et al., 2022), this intuitive way to enumerate all DAGs is computationally expensive and unrealistic.

### 3.2 A LOCAL ALGORITHM FOR FINDING POSSIBLE ADJUSTMENT SETS AND PROPENSITIES

Instead of searching globally for all possible DAGs, we adopt a novel framework called IDA (Maathuis et al., 2009; Fang, 2020), which can list all possible parent sets in CPDAG quickly. We further generalized the above theoretical results to MPDAGs with background knowledge added, and propose a local algorithm for finding possible adjustment sets and estimating corresponding possible propensities. As we will see later in Section 3.3, these counterfactual quantities can be sufficient to help us control for all possible counterfactual treatment effects used to assess counterfactual fairness.

Specifically, for three distinct vertices $X_i, X_j$ and $X_k$, if $X_i \to X_j \leftarrow X_k$ and $X_i$ is not adjacent to $X_k$ in $\mathcal{G}$, then the triplet $(X_i, X_j, X_k)$ is called a *v-structure* collided on $X_j$. Pearl (2000) have shown that two DAGs are equivalent if and only if they have the same skeleton and the same *v-structures*. Given a CPDAG $\mathcal{G}^*$ contains all DAGs equivalent to $\mathcal{G}$, let $\mathbf{S}(A)$ be a subset of $sib(A, \mathcal{G}^*)$, and $\mathcal{G}^*_{\mathbf{S}(A) \to A}$ denote a graph that is obtained from $\mathcal{G}^*$ by changing all undirected edges $\{Z - A, \forall Z \in \mathbf{S}(A)\}$ into the directed edges $\{Z \to A, \forall Z \in \mathbf{S}(A)\}$ and all of other undirected edges $\{Z - A, \forall Z \notin \mathbf{S}(A)\}$ into the directed edges with opposite direction $\{Z \leftarrow A, \forall Z \notin \mathbf{S}(A)\}$. We say $\mathbf{S}(A) \to A$ is a possible parent set of the sensitive attribute $A$ for $\mathcal{G}^*$, if there is a DAG $\mathcal{G}$ in the equivalence class $\mathcal{G}^*$ with the same directed edges adjacent to $A$ as $\mathcal{G}^*_{\mathbf{S}(A) \to A}$. Motivated by IDA (Maathuis et al., 2009), we show a sufficient and necessary condition for determining whether a set $\mathbf{S}(A) \subset \mathrm{sib}(A, \mathcal{G}^*)$ is a possible parent set of the sensitive attribute $A$ in below.

**Proposition 3.1.** *Given a CPDAG $\mathcal{G}^*$, a set $\mathbf{S}(A) \subset \mathrm{sib}(A, \mathcal{G}^*)$ is a possible parent set of the sensitive attribute $A$, if and only if there is no more v-structure in $\mathcal{G}^*_{\mathbf{S}(A) \to A}$ comparing $\mathcal{G}^*$.*

For MPDAG, as discussed in Fang (2020), a key difference compared with CPDAG is the possible generation of a directed triangular cycle (e.g., $A \to X_1 \to X_2 \to A$), when incorporating the background knowledge and using Meek's rule for orienting undirected edges adjacent to the sensitive attribute $A$. Motivated by such difference, in proposition 3.2, we generalize the above theoretical results to MPDAGs for determining possible parent sets of the sensitive attribute $A$.

---

**Algorithm 1:** A local algorithm for finding possible adjustment sets and estimating corresponding propensity model parameters of the sensitive attribute $A$ further using direct causal information.

---

**Input:** Sensitive attribute $A$, CPDAG $\mathcal{G}^*$, and consistent direct causal information set $\mathcal{B}_d$.

1  Construct the MPDAG $\mathcal{H}$ from $\mathcal{G}^*$ and $\mathcal{B}_d$ using Meek's rules;
2  Set $\mathcal{S}_A = \emptyset$ and $m = 1$;
3  **for** *each* $\mathbf{S}^{(m)} \subset \mathrm{sib}(A, \mathcal{G}^* \text{ or } \mathcal{H})$ *such that orienting* $\mathbf{S}^{(m)} \to A$ *and* $A \to \mathrm{sib}(A, \mathcal{G}^* \text{ or } \mathcal{H})$
   $\backslash \mathbf{S}^{(m)}$ *does not introduce any v-structure collided on* $A$ *or any directed triangle containing* $A$
   **do**
4      **for** *number of steps for training the possible propensity model on* $\mathbf{S}^{(m)}$ **do**
5          Sample a batch of units $\{(a_{m_k}, x_{m_k}|_{\mathbf{S}^{(m)}})\}_{k=1}^{K}$;
6          Update $\hat{\phi}^{(m)}$ by descending along the gradient $\nabla_{\hat{\phi}^{(m)}} \ell(\hat{\phi}^{(m)}; \mathbf{S}^{(m)})$;
7      **end**
8      $\mathcal{S}_A \leftarrow \mathcal{S}_A \cup \mathbf{S}^{(m)}$ and $m \leftarrow m + 1$;
9  **end**

**Output:** A set $\mathcal{S}_A$ of possible adjustment sets $\mathbf{S}^{(m)}$ and propensity model parameters $\hat{\phi}^{(m)}$.

---

**Proposition 3.2.** *Given an MPDAG $\mathcal{H}$, a set $\mathbf{S}(A) \subset \mathrm{sib}(A, \mathcal{H})$ is a possible parent set of $A$, if and only if there is no more direct triangle and v-structure in $\mathcal{H}_{\mathbf{S}(A) \to A}$ comparing $\mathcal{H}$.*

Empirically, propositions 3.1 and 3.2 can be implemented in a local manner as follows. For any vertex set $V' \subseteq V$, we define the *induced subgraph* of $\mathcal{G} = (V, E)$ over $V'$ by restricting the edges $E$ on the set of vertices $V'$. Therefore, the edge set $E'$ of the induced subgraph $\mathcal{G}' = (V', E')$ is defined as the subset of $E$ containing all edges with both endpoints in $V'$. Then proposition 3.1 is equivalent to check whether the induced subgraph of $\mathcal{G}^*$ over $\mathbf{S}(A)$ is complete, whereas proposition 3.2 is equivalent to checking whether the induced subgraph of $\mathcal{H}$ over $\mathbf{S}(A)$ is complete, as well as there does not exist $S \in \mathbf{S}(A)$ and $C \in \mathrm{adj}(A, \mathcal{H}) \backslash (\mathbf{S}(A) \cup pa(A, \mathcal{H}))$ such that $C \to S$. This provides a computationally convenient way to locally find the possible parent sets of the sensitive attribute $A$.

Next, to estimate the counterfactual quantity $\hat{Y}_{A \leftarrow a}$, i.e., the counterfactual outcome of the predictor $\hat{Y}$ when setting the value of the sensitive attribute $A$ to $a$. We propose to first estimate $P(A \mid pa(A))$, called propensity, for each possible DAG in the Markov equivalence class. Specifically, for each possible parent set $\mathbf{S}^{(m)} \in \mathcal{S}_A$ for $m = 1, \ldots, |\mathcal{S}_A|$, we regress the sensitive attribute $A$ using the observed variables $X$ restricted on $\mathbf{S}^{(m)}$, denoted as $X|_{\mathbf{S}^{(m)}}$. We train the corresponding propensity model $g(X|_{\mathbf{S}^{(m)}}; \hat{\phi}^{(m)})$ for estimating $P(A \mid \mathbf{S}^{(m)})$ by minimizing the cross-entropy loss

$$\ell(\hat{\phi}^{(m)}; \mathbf{S}^{(m)}) = -\frac{1}{N} \sum_{i=1}^{N} \left[ A_i \log g(x_i|_{\mathbf{S}^{(m)}}; \hat{\phi}^{(m)}) + (1 - A_i) \log \left( 1 - g(x_i|_{\mathbf{S}^{(m)}}; \hat{\phi}^{(m)}) \right) \right],$$

where $\hat{\phi}^{(m)}$ is the learned parameter of the propensity model, and $\hat{e}_i^{(m)} = g(x_i|_{\mathbf{S}^{(m)}}; \hat{\phi}^{(m)})$ is the estimated propensity of unit $i$ corresponding to the possible parent set $\mathbf{S}^{(m)}$ for $i = 1, \ldots, N$ and $m = 1, \ldots, |\mathcal{S}_A|$. We summarized the proposed local algorithm in Alg. 1, where the text in blue color represents the extra steps in implementation on MPDAG compared with CPDAG.

### 3.3 Quantifying and Bounding Counterfactual Fairness

We then aim to estimate and bound all possible counterfactual treatment effects of sensitive attribute $A$ on the predictor $\hat{Y}$, begin with the counterfactual independence theorem in Pearl et al. (2016).

**Lemma 3.3** (Counterfactual independence theorem, in Section 4.3.2 of Pearl et al. (2016)). *Given an ordered pair of variables $(A, \hat{Y})$ in a DAG $\mathcal{G}$, suppose a set of variables $Z$ satisfies the condition that no node in $Z$ is a descendant of $A$ and that $Z$ blocks every path between $A$ and $Y$ that contains an arrow into $A$. Then, for all $a$, the counterfactual $\hat{Y}_{A \leftarrow a}$ is conditionally independent of $A$ given $Z$*

$$P\left(\hat{Y}_{A \leftarrow a} \mid A, Z\right) = P\left(\hat{Y}_{A \leftarrow a} \mid Z\right).$$

---

**Algorithm 2:** A min-max optimization approach alternatively updating possible counterfactual treatment effect models and prediction model controlling the worse-case fairness violations.

---

**Input:** Sensitive attribute $A$, outcome of interest $Y$, and other observable attributes $X$, possible adjustment sets $\mathbf{S}^{(m)}$ and propensity model parameters $\hat{\phi}^{(m)}$ from Alg. 1.

1 **while** *stopping criteria is not satisfied* **do**
2    **for** $m = 1, \ldots, |\mathcal{S}_A|$ **do**
3       **for** *number of steps for training the possible counterfactual treatment effect model* **do**
4          Sample a batch of units $\{(a_{m_k}, x_{m_k}, y_{m_k})\}_{k=1}^{K}$;
5          Update $\hat{\psi}^{(m)}$ by descending along the gradient $\nabla_{\hat{\psi}^{(m)}} \ell(\hat{\psi}^{(m)}; \theta)$;
6       **end**
7       Compute possible counterfactual treatment effects $\hat{\tau}_i^{(m)} = h(x_i|_{\mathbf{S}^{(m)}}; \hat{\psi}^{(m)})$;
8    **end**
9    **for** *number of steps for training the prediction model* **do**
10       Sample a batch of units $\{(a_l, x_l, y_l)\}_{l=1}^{L}$;
11       Update $\theta$ by descending along the gradient of min-max loss $\nabla_\theta \ell(\theta; \hat{\psi}^{(1)}, \ldots, \hat{\psi}^{(|\mathcal{S}_A|)})$;
12    **end**
13 **end**

---

In order to identify and estimate the counterfactual outcomes of the predictor $\hat{Y}$, we consider $\hat{Y}$ as a new node in the causal graph. Despite the true graph is unknown, a key observation is that $\hat{Y}$ cannot be a parent node of the sensitive attribute $A$, i.e., $\hat{Y} \notin pa(A, \mathcal{G})$, since the sensitive attribute cannot be affected by the predictor. Notably, the parent set $pa(A, \mathcal{G})$ satisfies the above conditions in lemma 3.3, i.e., no node in $pa(A, \mathcal{G})$ is a descendant of $A$, as implied by the definition of $pa(A, \mathcal{G})$, and $pa(A, \mathcal{G})$ blocks every path between $A$ and $\hat{Y}$ that contains an arrow into $A$, since $pa(A, \mathcal{G})$ contains all nodes directed to $A$. This illustrates the sufficiency of adjusting the parent sets of the sensitive attribute $A$ for identifying the counterfactual treatment effect of $A$ on $\hat{Y}$, with a high-level conclusion that under the *parental Markov condition*, i.e., every variable is independent of all its non-descendants conditional on its parents. We formally state the identifiability result in the following.

**Proposition 3.4.** *Under the parental Markov condition and the consistency assumption that $\hat{Y}_{A \leftarrow a}$ is the same as $\hat{Y}$ under $A = a$, for any predictor $\hat{Y} \notin pa(A, \mathcal{G})$, we have*

$$P(\hat{Y}_{A \leftarrow a} = y | pa(A) = z) = P(\hat{Y}_{A \leftarrow a} = y | pa(A) = z, A = a) = P(\hat{Y} = y | pa(A) = z, A = a).$$

Then the counterfactual outcome of the sensitive attribute on the predictor is equivalent to

$$P(\hat{Y}_{A \leftarrow a} = y) = \sum_z \frac{P(\hat{Y} = y, A = a | pa(A) = z)}{P(A = a | pa(A) = z)} P(pa(A) = z),$$

where the conclusion follows from the proposition 3.4 and the total probability formula. Given $\hat{e}_i^{(m)} = g(x_i|_{\mathbf{S}^{(m)}}; \hat{\phi}^{(m)})$ as the estimates of the propensity $P(A = a | pa(A) = z)$ in Section 3.2, we build a counterfactual treatment effect model for each possible parent set $\mathbf{S}^{(m)}$ for $m = 1, \ldots, |\mathcal{S}_A|$, and train the counterfactual treatment effect model $h(x_i|_{\mathbf{S}^{(m)}}; \hat{\psi}^{(m)}) = \hat{\tau}_i^{(m)}$ to estimate $P(\hat{Y}_{A \leftarrow 1}(U) = y_i \mid X = x_i, A = a_i) - P(\hat{Y}_{A \leftarrow 0}(U) = y_i \mid X = x_i, A = a_i)$[1] by minimizing

$$\ell(\hat{\psi}^{(m)}; \theta) = \sum_{i=1}^{N} \left( \frac{A_i f_{A \leftarrow 1}(x_i; \theta)}{\hat{e}_i^{(m)}} - \frac{(1 - A_i) f_{A \leftarrow 0}(x_i; \theta)}{1 - \hat{e}_i^{(m)}} - h(x_i|_{\mathbf{S}^{(m)}}; \hat{\psi}^{(m)}) \right)^2,$$

where $f_{A \leftarrow 1}(x_i; \theta)$ represents the predicted outcome of unit $i$ using the outcome predictor $f(x_i; \theta) = \hat{Y}_i$ had the sensitive attribute taken the value of $A_i = 1$ (and other variables had taken the value of $X_{A_i \leftarrow 1}$). Remarkably, $h(x_i|_{\mathbf{S}^{(m)}}; \hat{\psi}^{(m)}) = \hat{\tau}_i^{(m)}$ aims to evaluate the counterfactual fairness of the predictor $\hat{Y}$, thus strictly depends on the form of the predictor $\hat{Y} = f(x; \theta)$.

---

[1] This parameter of interest would be unidentifiable when any node in $X$ lies on the causal path from $A$ to $\hat{Y}$. Therefore, for each possible DAG, we first adopt IDA framework (Maathuis et al., 2009) to locally determine the parent nodes of $A$, then only use these nodes as input to train the corresponding treatment effect model.

Table 2: Average RMSE and unfairness for synthetic datasets on the held-out test set.

| Noise = 1.5 | NODE = 10, EDGE = 20 | | NODE = 20, EDGE = 40 | | NODE = 30, EDGE = 60 | | NODE = 40, EDGE = 80 | |
|---|---|---|---|---|---|---|---|---|
| Method | RMSE ↓ | Unfairness ↓ | RMSE ↓ | Unfairness ↓ | RMSE ↓ | Unfairness ↓ | RMSE ↓ | Unfairness ↓ |
| Oracle | $0.757 \pm 0.349$ | $0.000 \pm 0.000$ | $0.579 \pm 0.245$ | $0.000 \pm 0.000$ | $0.571 \pm 0.194$ | $0.000 \pm 0.000$ | $0.578 \pm 0.200$ | $0.000 \pm 0.000$ |
| Full | $0.576 \pm 0.218$ | $0.195 \pm 0.232$ | $0.494 \pm 0.133$ | $0.095 \pm 0.128$ | $0.542 \pm 0.196$ | $0.063 \pm 0.083$ | $0.538 \pm 0.183$ | $0.067 \pm 0.113$ |
| Unaware | $0.587 \pm 0.219$ | $0.150 \pm 0.208$ | $0.498 \pm 0.134$ | $0.058 \pm 0.095$ | $0.544 \pm 0.196$ | $0.050 \pm 0.076$ | $0.540 \pm 0.183$ | $0.043 \pm 0.066$ |
| FairRelax | $0.653 \pm 0.256$ | $0.142 \pm 0.201$ | $0.586 \pm 0.217$ | $0.055 \pm 0.092$ | $0.603 \pm 0.241$ | $0.045 \pm 0.068$ | $0.611 \pm 0.254$ | $0.041 \pm 0.068$ |
| Fair | $0.747 \pm 0.293$ | $0.128 \pm 0.200$ | $0.627 \pm 0.223$ | $0.050 \pm 0.074$ | $0.661 \pm 0.263$ | $0.043 \pm 0.067$ | $0.630 \pm 0.292$ | $0.038 \pm 0.059$ |
| Ours | $\mathbf{0.623 \pm 0.210}$ | $\mathbf{0.119 \pm 0.175}$ | $\mathbf{0.561 \pm 0.126}$ | $\mathbf{0.049 \pm 0.073}$ | $\mathbf{0.597 \pm 0.185}$ | $\mathbf{0.037 \pm 0.054}$ | $\mathbf{0.606 \pm 0.178}$ | $\mathbf{0.036 \pm 0.054}$ |

| Noise = 2.5 | NODE = 10, EDGE = 20 | | NODE = 20, EDGE = 40 | | NODE = 30, EDGE = 60 | | NODE = 40, EDGE = 80 | |
|---|---|---|---|---|---|---|---|---|
| Method | RMSE ↓ | Unfairness ↓ | RMSE ↓ | Unfairness ↓ | RMSE ↓ | Unfairness ↓ | RMSE ↓ | Unfairness ↓ |
| Oracle | $0.729 \pm 0.344$ | $0.000 \pm 0.000$ | $0.874 \pm 0.625$ | $0.000 \pm 0.000$ | $0.801 \pm 0.497$ | $0.000 \pm 0.000$ | $0.820 \pm 0.472$ | $0.000 \pm 0.000$ |
| Full | $0.667 \pm 0.274$ | $0.185 \pm 0.189$ | $0.761 \pm 0.440$ | $0.150 \pm 0.425$ | $0.736 \pm 0.417$ | $0.075 \pm 0.087$ | $0.729 \pm 0.334$ | $0.110 \pm 0.183$ |
| Unaware | $0.674 \pm 0.276$ | $0.065 \pm 0.094$ | $0.772 \pm 0.457$ | $0.062 \pm 0.126$ | $0.737 \pm 0.417$ | $0.032 \pm 0.043$ | $0.733 \pm 0.336$ | $0.041 \pm 0.079$ |
| FairRelax | $0.738 \pm 0.283$ | $0.059 \pm 0.077$ | $0.898 \pm 0.600$ | $0.050 \pm 0.119$ | $0.831 \pm 0.487$ | $0.030 \pm 0.040$ | $0.791 \pm 0.410$ | $0.040 \pm 0.079$ |
| Fair | $0.774 \pm 0.274$ | $0.052 \pm 0.067$ | $0.937 \pm 0.642$ | $0.046 \pm 0.118$ | $0.891 \pm 0.550$ | $0.029 \pm 0.039$ | $0.816 \pm 0.411$ | $0.039 \pm 0.079$ |
| Ours | $\mathbf{0.719 \pm 0.280}$ | $\mathbf{0.049 \pm 0.073}$ | $\mathbf{0.857 \pm 0.466}$ | $\mathbf{0.045 \pm 0.090}$ | $\mathbf{0.823 \pm 0.413}$ | $\mathbf{0.023 \pm 0.031}$ | $\mathbf{0.788 \pm 0.334}$ | $\mathbf{0.038 \pm 0.070}$ |

## 3.4 MIN-MAX JOINT LEARNING APPROACH

We now aim to train a predictor to satisfy counterfactual fairness. In contrast to previous variable selection methods based on causal discovery (Zuo et al., 2022), the proposed learning approach can effectively exploit *all* variables to make predictions, which improves the prediction accuracy.

Since the parent set of the sensitive attribute in the true DAG is unknown, we propose a min-max learning approach to control for the worst-case counterfactual fairness violations of the predictor. Specifically, given all possible individual causal effects $\hat{\tau}_i^{(m)}$ of the sensitive attribute $A$ on the predictor $\hat{Y}$ in Section 3.3, the prediction model $\hat{Y} = f(x; \theta)$ is trained by minimizing the average prediction error with the worst-case violations of counterfactual fairness as a penalty term

$$\min_\theta \ell(\theta; \hat{\psi}^{(1)}, \ldots, \hat{\psi}^{(|\mathcal{S}_A|)}) = \sum_{i=1}^N (Y_i - f(x_i; \theta))^2 + \gamma \cdot \max_m \sum_{i=1}^N \xi_i^{(m)},$$

$$\text{s.t. } \hat{\tau}_i^{(m)} \leq C + \xi_i^{(m)}, \quad i = 1, \ldots, N, \quad m = 1, \ldots, |\mathcal{S}_A|,$$

$$\hat{\tau}_i^{(m)} \geq -C - \xi_i^{(m)}, \quad i = 1, \ldots, N, \quad m = 1, \ldots, |\mathcal{S}_A|,$$

$$\xi_i^{(m)} \geq 0, \quad i = 1, \ldots, N, \quad m = 1, \ldots, |\mathcal{S}_A|,$$

which is a convex optimization problem when $\hat{\tau}_i^{(m)} = h(x_i; \hat{\psi}^{(m)})$ is linear. It is equivalent to

$$\min_\theta \tilde{\ell}(\theta) = \sum_{i=1}^N (Y_i - f(x_i; \theta))^2 + \lambda \cdot \max_m \sum_{i=1}^N \left[ (-C - \hat{\tau}_i^{(m)})_+ + (\hat{\tau}_i^{(m)} - C)_+ \right],$$

where $\gamma$ and $\lambda$ are hyper-parameters for trade-off between prediction accuracy and counterfactual fairness. Since achieving strict counterfactual fairness for all individuals, i.e., having zero individual causal effects of sensitive attribute on the predictor, is usually unrealistic and would come at the cost of much prediction accuracy, we introduce a slack variable $\xi_i^{(m)}$ for each individual and a pre-specified threshold $C$, which penalizes the loss when the estimated individual causal effect $|\hat{\tau}_i^{(m)}| > C$.

Note that when implementing the proposed min-max optimization approach, the possible counterfactual treatment effect models in Section 3.3 and the prediction model controlling for worse-case fairness violations in Section 3.4 should be updated *alternatively*, which can be viewed as an iterative process of *counterfactual evaluation* and *policy improvement* of the prediction model. We summarized the whole min-max optimization algorithm in Alg. 2.

## 4 EXPERIMENTS

In this section, both synthetic and real-world experiments are conducted to evaluate the prediction accuracy and fairness of our approach. The root mean squared error (RMSE) between $Y$ and $\hat{Y}$ is used to measure the prediction performance, and the RMSE between $\hat{Y}_{A \leftarrow a}$ and $\hat{Y}_{A \leftarrow a'}$ is used to measure the violation of the counterfactual fairness, named "unfairness".

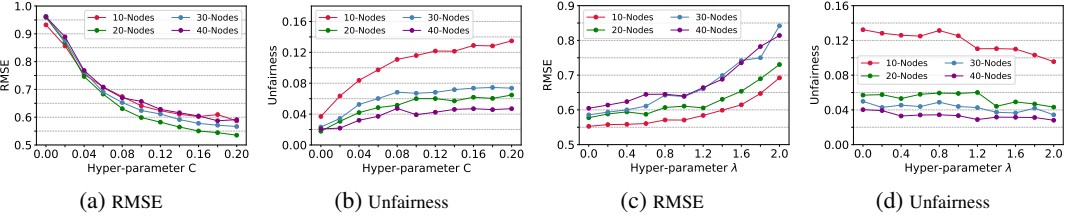

Figure 2: Performance under varying hyper-parameters C and $\lambda$ on RMSE and unfairness.

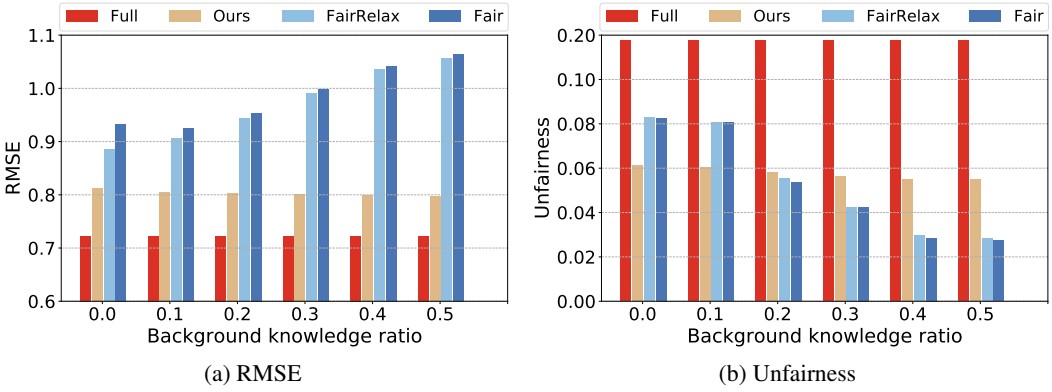

Figure 3: RMSE and unfairness performance under varying background knowledge ratio.

**Baselines.** We consider five baseline prediction models: (1) **Full** uses all attributes, (2) **Unaware** uses all attributes except the sensitive attribute, (3) **Oracle** uses all attributes that are non-descendants of the sensitive attribute given the ground-truth DAG, (4) **FairRelax** uses all definite non-descendants and possible descendants of the sensitive attribute in a CPDAG (or an MPDAG), and (5) **Fair** uses all definite non-descendants of the sensitive attribute in a CPDAG (or an MPDAG).

Table 3: Average precision and recall for finding the adjustment sets in MPDAG.

| Background Knowledge | 0% | 10% | 20% | 30% | 40% | 50% |
|---|---|---|---|---|---|---|
| Precision ↑ | $0.438 \pm 0.489$ | $0.438 \pm 0.489$ | $0.466 \pm 0.492$ | $0.580 \pm 0.486$ | $0.642 \pm 0.471$ | $0.742 \pm 0.437$ |
| Recall ↑ | $0.265 \pm 0.353$ | $0.265 \pm 0.353$ | $0.274 \pm 0.350$ | $0.329 \pm 0.355$ | $0.353 \pm 0.347$ | $0.400 \pm 0.346$ |

**Synthetic Study.** Synthetic data are generated from a linear structural equation model based on a ground-truth DAG. Specifically, we first randomly generate a DAG with $d$ nodes and $2d$ directed edges according to the Erdős-Rényi (ER) model with $d \in \{10, 20, 30, 40\}$ in our experiment. Following the previous study (Zuo et al., 2022), the path coefficients $\beta_{jk}$ of directed edges $X_j \rightarrow X_k$ are sampled from a Uniform($[-2, -0.5] \cup [0.5, 2]$) distribution. The data are generated using $X_k = \sum_{X_j \in pa(X_k)} \beta_{jk} X_j + \epsilon_i, i = 1, \ldots, n$, where $pa(X_k)$ represents the parent nodes of $X_k$, noise $\epsilon_i \sim N(0, \gamma)$ with $\gamma \in \{1.5, 2.5\}$, and $n$ is the sample size, which is set to 1,000 in our experiment. We next use the PC algorithm in the causal-learn package to learn a CPDAG. Then we randomly select two nodes as the outcome $Y$ and the sensitive attribute $A$, respectively. We sample $A$ from a Binomial($[0,1]$) distribution with probability $\sigma(\sum_{X_j \in pa(A)} \beta_{jA} X_j + \epsilon_i)$, where $\sigma(\cdot)$ denotes the sigmoid function. The proportion of training data and test data are set to 0.8 and 0.2, respectively.

**Performance Comparison.** Table 2 shows the results of baselines and our approach. First, **Full** and **Unaware** perform better on RMSE, while **Fair**, **FairRelax**, and our approach have a significant advantage on unfairness. Remarkably, our approach outperforms **Fair** and **FairRelax** in all scenarios on both RMSE and unfairness metrics, because the proposed method makes predictions with all attributes and controls unfairness by the adjustment sets, whereas **Fair** and **FairRelax** can hardly find the true descendants of the sensitive attribute when the learned CPDAG is not accurate. In addition, Figure 2 shows the change in RMSE and unfairness as $C$ and $\lambda$ increase. When $C$ is increasing, RMSE is decreasing significantly, while unfairness is increasing. Because the larger $C$ is, the looser

Table 4: Real-world experiment results.

|  | Full | Unaware | FairRelax | Fair | Ours |
|---|---|---|---|---|---|
| RMSE | $0.502 \pm 0.041$ | $0.502 \pm 0.042$ | $0.503 \pm 0.041$ | $0.503 \pm 0.041$ | $\mathbf{0.491 \pm 0.040}$ |
| Unfairness | $0.088 \pm 0.024$ | $0.031 \pm 0.058$ | $0.029 \pm 0.023$ | $0.029 \pm 0.023$ | $\mathbf{0.024 \pm 0.018}$ |

(a) Full    (b) Unaware    (c) Fair    (d) FairRelax    (e) Ours

Figure 4: Density plot of the predicted $\hat{Y}_{A \leftarrow a}(x)$ and $\hat{Y}_{A \leftarrow a'}(x)$ in real-world data.

the control of causal effects, which is beneficial for prediction performance but hurts fairness. Similar arguments hold for $\lambda$, where a larger $\lambda$ will increase the cost of fairness violations in the optimization problem, thus benefiting fairness but hurting prediction performance.

**MPDAG with Background Knowledge**. After obtaining a CPDAG, we randomly select a certain percentage of the directed edges in the true DAG as background knowledge and impose it on the already learned CPDAG. For example, if $A \rightarrow B$ is selected from the true DAG, we add this directed edge to the learned CPDAG regardless of the original relationship between A and B in the CPDAG to obtain an MPDAG, and then adjust the MPDAG according to the Meek's rule. Figure 3 shows the effect of background knowledge on performance. As the background knowledge increases, the RMSE of **Fair** and **FairRelax** increases and the unfairness decreases significantly, because more background knowledge force **Fair** and **FairRelax** to have fewer nodes to make predictions. For our approach, both prediction and unfairness performance become slightly better, which is attributed to the more accurate identification of the possible adjustment sets. Our approach is less sensitive to the increase in background knowledge compared with **Fair** and **FairRelax**, since we do not directly exploit the graph information and select the nodes for prediction. In addition, Table 3 reports the change of precision and recall for finding adjustment sets with increasing background knowledge.

**Case Study.** The Open University Learning Analytics Dataset (OULAD) dataset (Kuzilek et al., 2017) is used for the real-world experiment. The data attributes includes demographic information about the students such as gender, age, education level, disability and other attributes as well as their final grades. This dataset contains 32,593 students and 11 attributes. We treat disability as the sensitive attribute and binarize the final grades as the outcome of interest. First, we learn a CPDAG from the raw data using the PC algorithm in the causal-learn package and obtain an MPDAG with the background knowledge that sex can not be caused by other attributes. Second, we randomly generate a DAG as the ground-truth from the learned MPDAG. We then divide the data into 100 random batches, and for each batch, a new MPDAG is obtained from a similar way. The path coefficients are determined based on linear regression and treat the residual of the regression as noise. The subsequent steps are the same as in the synthetic study. The experiment results are shown in Table 4, with density plots in Figure 4, and our approach outperform baselines in both prediction performance and fairness.

## 5   CONCLUSION

This paper aims to achieve counterfactual fairness from observational data when the causal graph is unknown or partially known. Interestingly, we show it is actually possible to achieve counterfactual fairness even using the descendants of the sensitive attribute for prediction, by carefully control the possible counterfactual effects of the sensitive attribute. We propose a general min-max optimization framework to achieve counterfactual fairness that is easy applicable to CPDAGs and maximally oriented PDAGs (MPDAGs) with the added background knowledge. Similar to previous studies, one limitation of our approach is due to the proposed approach relying on a CPDAG given by the causal discovery algorithm and estimations of the propensities, which may lead to mild violations of counterfactual fairness by the algorithm when the CPDAG or estimates are inaccurate. Another possible limitation, which also serves as a future research direction, is to achieve counterfactual fairness in the presence of hidden variables with partially known DAGs.

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
