# OpenReview forum: "Counterfactual Fairness from Partially DAGs: A General Min-Max Optimization Framework"
_ICLR.cc/2024/Conference — Submitted to ICLR 2024_

### Official Review · Reviewer_yEED · 2023-10-17

**Soundness:** 2 fair
**Presentation:** 3 good
**Contribution:** 3 good
**Rating:** 6
**Confidence:** 4

**Summary:**

The authors propose to learn a fair predictive model under counterfactual fairness constraint only with CPDAG or MPDAG. To achieve this, they employ a technique for enumerating the possible adjustment variables, construct a propensity model for each possible adjustment variable set, estimate causal effects, and train a fair predictive model. Overall, the paper is well written and addresses an important problem. Since I have several concerns, I will hold my overall rating.

**Strengths:**

- Achieving the causality-based fairness without complete prior knowledge about the causal graph is a very important topic.

- The idea of enumerating possible adjustment variable sets is reasonable.

**Weaknesses:**

(A) The motivation of introducing IDA algorithm seems unclear

To enumerate possible adjustment variable sets, the authors employ the existing IDA algorithm, which aims to find adjustment variables from the **siblings** of sensitive feature $A$. However, I could not understand why it is sufficient to focus on the siblings. I understand that it is easier to determine whether a variable is parent of $A$ if it is a sibling of $A$ (because of the property of v-structure). However, I could not understand why all possible confounder can be enumerated with the IDA algorithm.

- Could you elaborate why employing IDA algorithm is sufficient? I believe that if it fails to output possible confounders, the resulting propensity score model cannot eliminate the confounding bias.

(B) Clarity issues

- I could not understand why the authors present the formula of $P( \hat{Y}_{A \leftarrow a} = y)$ in Section 3.3. Is it used in the proposed method? If so, it is computationally demanding since there are so many possible values of $pa(A) = z$ if $z$ is high-dimensional.

- Please clearly state that prediction $\hat{Y}$ is regarded as a node (an endogenous variable) in the causal graph in Section 2.2. Otherwise, it is difficult to understand the problem setting. Is observed outcome $Y$ also a node in the causal graph, right?

**Questions:**

- Could you elaborate why employing IDA algorithm is sufficient? I believe that if it fails to output possible confounders, the resulting propensity score model cannot eliminate the confounding bias.

---

### Official Review · Reviewer_xASm · 2023-10-25

**Soundness:** 1 poor
**Presentation:** 3 good
**Contribution:** 2 fair
**Rating:** 3
**Confidence:** 4

**Summary:**

The paper proposes a novel max-min approach that combines causal discovery with counterfactual treatment effect estimation to learn a counterfactually fair predictor. The basic idea is to iterate over all DAGs in a learned Markov equivalence class and penalize the training objective with the corresponding worst-case counterfactual effect, jointly while updating the parameters of the predictor. The method is evaluated on both simulated and synthetic data.

**Strengths:**

- The paper targets an important problem at the intersection of causality and algorithmic fairness
- It is well written, and the main method is intuitive
- The experimental results are promising

**Weaknesses:**

I have two major concerns, which keep me from accepting the paper in its current state.

1) Identifiability of counterfactuals. The paper claims to achieve (point) identification of counterfactual predictions. However, I believe point identification is only possible here because, in Proposition 3.4., the authors do not condition on the descendants of the sensitive attribute $A$. This is in strong contrast to the main claim of the paper to use **all** observed variables for counterfactual fairness, not just the non-descendants. Furthermore, it is inconsistent with the proposed counterfactual estimator on page 6., in which I believe the authors do include descendants of $A$. In general, estimating counterfactual predictions would require (at least implicitly) estimating the counterfactual descendants of $A$ (i.e., causal queries of the form $\mathbb{P}(M_{A \to a^\prime} \mid A = a, M = m)$, where $M$ are descendants of $A$. The identifiability theory of such causal queries is well-developed, and **counterfactual (point) identification is impossible** without imposing additional assumptions (Plecko and Bareinboim, 2022). Looking forward, I believe the authors have three options to improve the paper: (i) Imposing strong parametric assumptions on the data-generating process that enable counterfactual identification (and I think it is necessary that these assumptions are spelled out explicitly). (ii) Using a partial-identification approach, as done by Wu et al. (2019) for discrete variables. However, Melnychuk et al. (2023) showed that, for continuous variables, not even partial counterfactual identification is possible without imposing stronger assumptions. (iii) Choosing a weaker/less individualized (but identifiable) notion of fairness, such as interventional fairness (Kilbertus et al. 2017).

2) Min-max approach for achieving counterfactual fairness. The authors motivate their approach with the toy example in Figure 1. However, if I understand correctly, the proposed approach would coincide here with the FAIR method, as the worst-case DAG would include both $X_1$ and $X_2$ as descendants. More generally: Why does the min-max approach proposed in the paper not coincide with the FAIR method that excludes all (possible) descendents of $A$? I.e., why does the worst-case DAG not always include all (possible) descendants as actual descendants? I could imagine that this would induce certain conditional independences that would violate the learned Markov equivalence class. But I think it is important that the authors provide some intuition here, either via a theoretical result or another toy example, that shows that the proposed max-min approach is not equal to FAIR.

**Questions:**

Can you provide a toy example, that shows that the proposed max-min approach is not equal to FAIR?

---

### Official Review · Reviewer_MxRt · 2023-10-30

**Soundness:** 3 good
**Presentation:** 2 fair
**Contribution:** 2 fair
**Rating:** 5
**Confidence:** 1

**Summary:**

For counterfactual fairness (CF) by Kusner et al. (2017), we need knowledge of the full causal graph. This paper considers CF under a partially known causal graph, or PDAG for partially directed acyclic graph. In particular, it develops a min-max algorithm to ensure a counterfactually fair algorithm under a PDAG. Unlike previous work on PDAGs for CF, this paper uses all available attributes for learning the predictor. It does so by defining a local search algorithm via the IDA framework. The paper's approach allows to learn counterfactually fair predictors with high accuracy.

**Strengths:**

S1: The paper addresses a key shortcoming of counterfactual fairness: partial knowledge of the causal graph.

S2: The paper proposes a new algorithm that, based on the cited related works, is more efficient then the state of the art.

**Weaknesses:**

W1: I find it confusing that the paper at parts refers (e..g., at the start of Sec. 3.4) to its approach as a variable selection method. While I understand this phrasing for the previous works in which those algorithms only considers a subset of the available variables based on their relationship to the protected attribute $A$, the proposed algorithm considers all attributes, right? Under a variable selection problem, I would expect the algorithm to return a subset of attributes that satisfy counterfactual fairness the most, which is not the case in the experiments.

W2: It would help the paper to be more explicit about the limitations of the algorithm. For fairness, in particular, would it be able to handle more than one protected attribute? This is not discussed.

**Questions:**

Q1: How does this work compare the Kusner et al. (2017)'s companion paper, When Worlds Collide: Integrating Different Counterfactual Assumptions in Fairness by the same authors and also from NIPS 2017? The paper explores CF under multiple worldviews (i.e. DAGs) for a given context. It also shows how CF can be balanced across worlds while keeping model performance.

---

### Meta-Review · Area_Chair_6pPD · 2023-12-10

**Metareview:**

The paper proposes a min-max optimization framework enforce counterfactual fairness in a classifier, when the knowledge on the underlying causal graph (capturing the data generation process) is limited. The paper investigates an important problem that hinders (among others) the practical application counterfactual fair classification. However, the reviewers pointed out several major issues that were unfortunately address in the rebuttal period. I thus encourage the authors to improved their paper based on the received reviewers' comments.

**Justification For Why Not Higher Score:**

There are major issues (including identifiability of the counterfactuals, optimality of the approach, etc) with the paper that unfortunately have not been clarified, as no rebuttal was provided by the authors.

**Justification For Why Not Lower Score:**

N/A

---

### Decision · Program_Chairs · 2024-01-16

Reject